# Structural basis for human TRPC5 channel inhibition by two distinct inhibitors

**Kangcheng Song[1†], Miao Wei[1†], Wenjun Guo[1], Li Quan[1], Yunlu Kang[1], Jing-Xiang Wu[1,2,3], Lei Chen[1,2,3]\***

[1]State Key Laboratory of Membrane Biology, College of Future Technology, Institute of Molecular Medicine, Beijing Key Laboratory of Cardiometabolic Molecular Medicine, Peking University, Beijing, China; [2]Peking-Tsinghua Center for Life Sciences, Peking University, Beijing, China; [3]Academy for Advanced Interdisciplinary Studies, Peking University, Beijing, China

**Abstract** TRPC5 channel is a nonselective cation channel that participates in diverse physiological processes. TRPC5 inhibitors show promise in the treatment of anxiety disorder, depression, and kidney disease. However, the binding sites and inhibitory mechanism of TRPC5 inhibitors remain elusive. Here, we present the cryo-EM structures of human TRPC5 in complex with two distinct inhibitors, namely clemizole and HC-070, to the resolution of 2.7 Å. The structures reveal that clemizole binds inside the voltage sensor-like domain of each subunit. In contrast, HC-070 is wedged between adjacent subunits and replaces the glycerol group of a putative diacylglycerol molecule near the extracellular side. Moreover, we found mutations in the inhibitor binding pockets altered the potency of inhibitors. These structures suggest that both clemizole and HC-070 exert the inhibitory functions by stabilizing the ion channel in a nonconductive closed state. These results pave the way for further design and optimization of inhibitors targeting human TRPC5.

**\*For correspondence:**
chenlei2016@pku.edu.cn

[†]These authors contributed equally to this work

**Competing interests:** The authors declare that no competing interests exist.

## Introduction

The mammalian transient receptor potential canonical (TRPC) channels are $Ca^{2+}$-permeable nonselective cation channels that belong to the transient receptor potential (TRP) channel superfamily (*Clapham, 2003*). Among all the subfamilies of TRP channels, TRPC channels share the closest homology to the *Drosophila* TRP channel, the first TRP channel cloned (*Montell and Rubin, 1989*). The TRPC subfamily is formed by seven channels in mammals, TRPC1-TRPC7, while TRPC2 is a pseudogene in human. According to the primary amino acid sequences, TRPC channels can be divided into TRPC1, TRPC2, TRPC3/6/7, and TRPC4/5 subgroups (*Montell et al., 2002*). Among them, TRPC5 is mainly expressed in the brain and to a lesser extent in the liver, kidney, testis, and pancreas (*Philipp et al., 1998*; *Sossey-Alaoui et al., 1999*; *Uhlén et al., 2015*). It can form homotetrameric channels or heterotetrameric channels with TRPC4 and/or TRPC1 (*Chung et al., 2007*; *Chung et al., 2006*). TRPC5 activation leads to the depolarization of cell membrane and an increase of intracellular $Ca^{2+}$ level ($[Ca^{2+}]_i$). TRPC5 mediates diverse physiological processes and is implicated in many disease conditions in human such as fear, anxiety, depression, and progressive kidney disease (*Riccio et al., 2009*; *Schaldecker et al., 2013*).

TRPC5 channel is regulated by several physiological mechanisms, including cell surface receptors, calcium stores, redox status, and calcium. TRPC5 can be activated by receptor-activated $G_{q/11}$-PLC pathway upon the dissociation of $Na^+/H^+$ exchanger regulatory factor and activation of $IP_3$ receptors (*Kanki et al., 2001*; *Schaefer et al., 2000*; *Storch et al., 2017*), or through the $G_{i/o}$ pathway (*Jeon et al., 2012*). By interacting with STIM1, TRPC5 functions as a store-operated channel (*Asanov et al., 2015*; *Zeng et al., 2004*). Extracellular application of reduced thioredoxin or

reducing agents enhances TRPC5 activity (*Xu et al., 2008*). TRPC5 is also regulated by $[Ca^{2+}]_i$ in a concentration-dependent bell-shaped manner (*Blair et al., 2009*; *Gross et al., 2009*; *Ordaz et al., 2005*; *Zeng et al., 2004*), and extracellular $Ca^{2+}$ ($[Ca^{2+}]_o$) at concentration higher than 5 mM robustly activates TRPC5 (*Schaefer et al., 2000*; *Zeng et al., 2004*).

Several available pharmacological tool compounds permit the identification of TRPC5 as putative drug targets for several diseases in human (*Minard et al., 2018*; *Sharma and Hopkins, 2019*; *Wang et al., 2020*). (-)-Englerin A (EA), a natural product obtained from *Phyllanthus engleri*, can selectively activate TRPC4/5 and therefore inhibit tumor growth (*Akbulut et al., 2015*; *Carson et al., 2015*), suggesting activation of TRPC4/5 might be a strategy to cure certain types of cancer. Moreover, TRPC5 inhibitors showed promising results for the treatment of central nervous system diseases and focal segmental glomerulosclerosis in animal models (*Just et al., 2018*; *Yang et al., 2015*; *Zhou et al., 2017*). A TRPC4/5 inhibitor developed by Hydra and Boehringer Ingelheim is currently in clinical trial for the treatment of anxiety disorder and depression (*Wulff et al., 2019*). Another TRPC5 inhibitor GFB-887 developed by Goldfinch Bio is in phase 1 clinical trial for the treatment of kidney disease (NCT number: NCT03970122).

Among TRPC5 inhibitors, clemizole (CMZ) and HC-070 represent two distinct classes in terms of chemical structures. CMZ is a benzimidazole-derived H1 antagonist that can inhibit TRPC5. M084 and AC1903 are also TRPC5 inhibitors that belong to the CMZ class (*Richter et al., 2014*; *Zhou et al., 2017*; *Zhu et al., 2015*). In contrast, HC-070 is a methylxanthine derivative that inhibits TRPC4/5 with high potency (*Just et al., 2018*). TRPC5 inhibitor Pico145 (HC-608) and activator AM237 belong to this class (*Just et al., 2018*; *Minard et al., 2019*). The distinct sizes, shapes, and chemical characteristics between CMZ and HC-070 type inhibitors suggest that they bind human TRPC5 (hTRPC5) at different sites. However, these sites remain enigmatic due to the lack of reliable structural information, which greatly impedes further structure-based compound optimizations. To understand the binding and inhibitory mechanism of these two types of inhibitors, we embarked on structural studies of hTRPC5 in complex with CMZ or HC-070.

## Results

### Structure of hTRPC5 in apo state and in complex with inhibitors

We show that extracellular calcium and EA can robustly elevate intracellular calcium concentration through hTRPC5 channel activation (*Figure 1—figure supplement 1A–C*), as reported previously (*Just et al., 2018*; *Schaefer et al., 2000*; *Zeng et al., 2004*). Moreover, CMZ and HC-070 effectively inhibited the extracellular calcium-induced intracellular calcium increase in a dose-dependent manner (*Figure 1A–B*). To reveal the binding sites of EA, CMZ, and HC-070 on hTRPC5, we expressed hTRPC5 in HEK293F cells for structural studies. It is previously reported that the C-terminal truncation of mTRPC increased expression level and retained its pharmacology profile (*Duan et al., 2019*). Similarly, we found the C-terminal truncation of hTRPC5 yielded a construct (hTRPC5$_{1-764}$) that can be activated by EA and extracellular calcium and can be inhibited by HC-070 and CMZ (*Figure 1—figure supplement 1A–C*). This construct showed a higher expression level of the tetrameric channel in comparison with the full-length wild-type (WT) hTRPC5 (*Figure 1—figure supplement 1D*). Therefore, we used hTRPC5$_{1-764}$ protein for structural studies. Tetrameric hTRPC5$_{1-764}$ channel was purified in detergent micelles (*Figure 1—figure supplement 1E–F*). To obtain the EA-bound, CMZ-bound, and the HC-070-bound TRPC5 structures, we added EA, CMZ, or HC-070 to the protein for cryo-EM sample preparation. For the cryo-EM sample in the presence of EA, we obtained a reconstruction at the resolution of 3.0 Å (*Figure 1—figure supplements 2–3*, *Table 1*). Unfortunately, we could not locate the density of EA, indicating EA was not bound on the protein, probably due to reduced affinity of EA toward purified TRPC5 protein in the current condition. Therefore, we refer this structure as the 'apo' state of hTRPC5, which closely resembles previously published 'apo' mouse TRPC5 structure (PDB ID: 6AEI) (*Duan et al., 2019*) with root-mean-square deviation (RMSD) of 1.844 Å. In addition, we obtained maps of hTRPC5 in CMZ-bound state and HC-070-bound state to the resolution of 2.7 and 2.7 Å, respectively (*Figure 1*, *Figure 1—figure supplements 4–7*, *Table 1*), which unambiguously revealed the ligand densities of CMZ and HC-070 described as below (*Figure 1—figure supplement 5E–F*, *Figure 1—figure supplement 7E–F*).

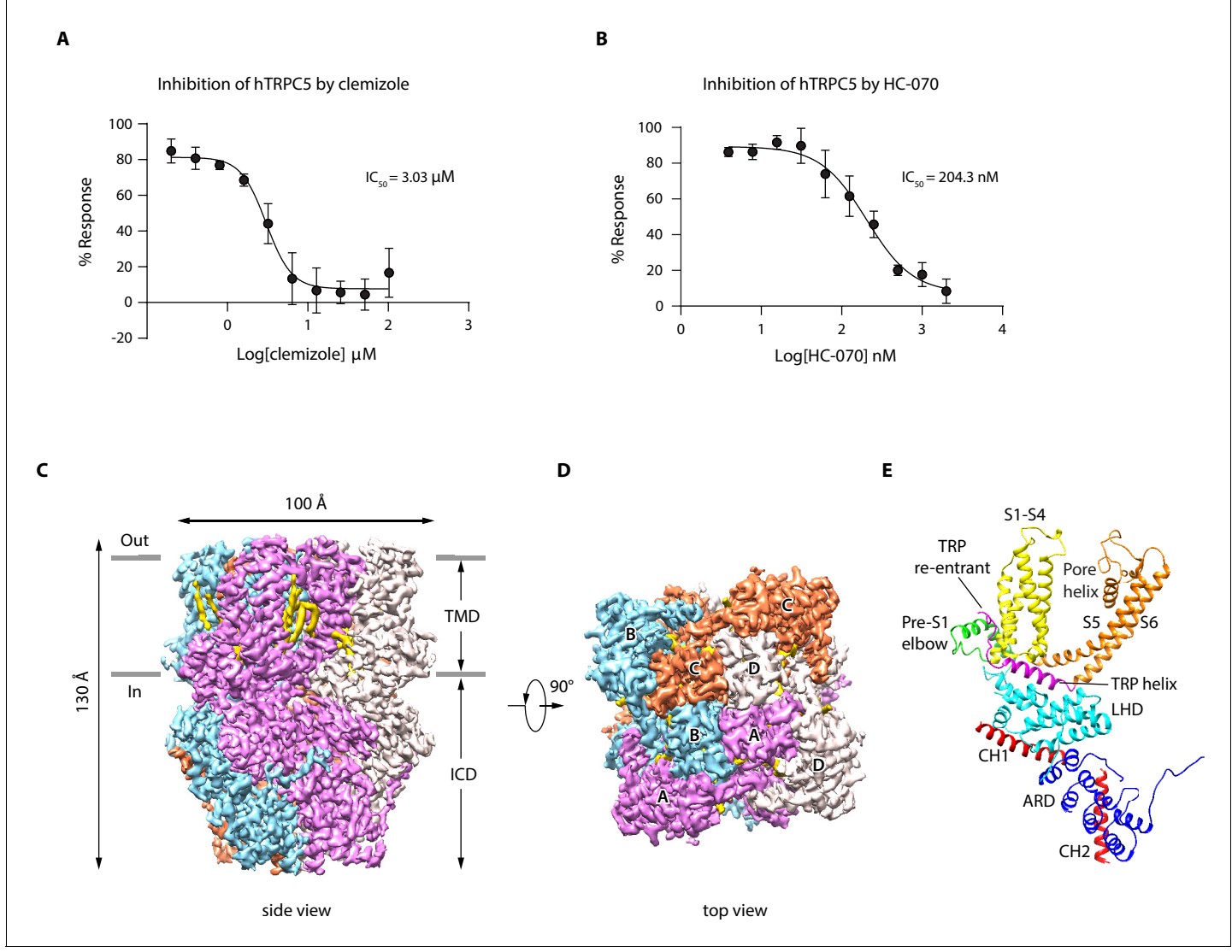

**Figure 1.** Overall structure of human TRPC5 (hTRPC5). (A, B) Inhibitory effect of clemizole (CMZ) (A) and HC-070 (B) on the extracellular calcium-induced intracellular calcium increase of cells with wild-type hTRPC5 over-expression measured by FLIPR calcium assay (data are shown as means ± standard error, n = 3 independent experiments). (C, D) Cryo-EM density maps of CMZ-bound hTRPC5 shown in side view (C) and top view (D). Subunits A, B, C, and D are colored in purple, light blue, orange, and gray, respectively. Lipids are colored in gold. The approximate boundary of the cell membrane is indicated by gray lines. TMD, transmembrane domain; ICD, intracellular cytosolic domain. (E) Ribbon diagram of a single subunit with secondary structure elements represented in different colors. ARD, ankyrin repeats domain; LHD, linker-helix domain; CH1, C-terminal helix 1; CH2, C-terminal helix 2.

The online version of this article includes the following source data and figure supplement(s) for figure 1:

**Source data 1.** Inhibition of WT hTRPC5 by clemizole or by HC-070.

**Figure supplement 1.** Biochemical and functional characterization of human TRPC5 (hTRPC5) constructs.

**Figure supplement 2.** Cryo-EM image analysis of apo human TRPC5 (hTRPC5).

**Figure supplement 3.** Cryo-EM map of apo human TRPC5 (hTRPC5).

**Figure supplement 4.** Cryo-EM image analysis of clemizole (CMZ)-bound human TRPC5 (hTRPC5).

**Figure supplement 5.** Cryo-EM map of clemizole (CMZ)-bound human TRPC5 (hTRPC5).

**Figure supplement 6.** Cryo-EM image analysis of HC-070-bound human TRPC5 (hTRPC5).

**Figure supplement 7.** Cryo-EM map of HC-070-bound human TRPC5 (hTRPC5).

**Table 1.** Cryo-EM data collection, refinement, and validation statistics.

| | CMZ-bound hTRPC5 | HC-070-bound hTRPC5 | apo_hTRPC5 |
|---|---|---|---|
| PDB ID | 7D4P | 7D4Q | 7E4T |
| EMDB ID | EMD-30575 | EMD-30576 | EMD-30987 |
| *Data collection and processing* | | | |
| Magnification | 130,000× | 165,000× | 130,000× |
| Voltage (kV) | 300 | 300 | 300 |
| Electron exposure ($e^-/Å^2$) | 50 | 50 | 50 |
| Defocus range (μm) | −1.5 to −2.0 | −1.5 to −2.0 | −1.5 to −2.0 |
| Pixel size (Å) | 1.045 | 0.821 | 1.045 |
| Symmetry imposed | *C4* | *C4* | *C4* |
| Initial particle images (no.) | 354,312 | 274,189 | 64,918 |
| Final particle images (no.) | 90,357 | 114,211 | 29,156 |
| Map resolution (Å) | 2.7 | 2.7 | 3.0 |
| Fourier shell correlation threshold | 0.143 | 0.143 | 0.143 |
| Map resolution range (Å) | 250–2.7 | 250–2.7 | 250–3.0 |
| *Refinement* | | | |
| Initial model used (PDB code) | 5Z96 | 5Z96 | 5Z96 |
| Model resolution (Å) | 2.7 | 2.7 | 3.0 |
| Fourier shell correlation threshold | 0.143 | 0.143 | 0.143 |
| Model resolution range (Å) | 250–2.7 | 250–2.7 | 250–3.0 |
| Map sharpening B factor ($Å^2$) | −119.0 | −118.1 | −119.0 |
| Model composition | | | |
| Nonhydrogen atoms | | 22,388 | 22,276 |
| Protein residues | 22,472 | 2672 | 2656 |
| Ligands | 2680 | 24 | 24 |
| | 28 | | |
| B factors ($Å^2$) | | | |
| Protein | 95.86 | 120.29 | 140.20 |
| ligand | 91.34 | 124.44 | 133.98 |
| Root-mean-square deviations | | | |
| Bond lengths (Å) | 0.004 | 0.004 | 0.004 |
| Bond angles (°) | 0.914 | 0.918 | 0.906 |
| Validation | | | |
| Validation | 1.21 | 1.32 | 1.30 |
| MolProbity score | 4.33 | 5.80 | 5.52 |
| Clashscore | 0.67 | 0.67 | 1.01 |
| Poor rotamers (%) | | | |
| Ramachandran plot | | | |
| Favored (%) | 98.78 | 98.63 | 98.93 |
| Allowed (%) | 1.22 | 1.37 | 1.07 |
| Disallowed (%) | 0.00 | 0.00 | 0.00 |

The overall architecture of inhibitor-bound hTRPC5 is similar to apo state, occupying 100 Å ×100 Å × 130 Å in three-dimensional (3D) space (*Figure 1C–D*). The fourfold symmetric channel has two-layer architecture, composed of the intracellular cytosolic domain (ICD) layer and the transmembrane domain (TMD) layer (*Figure 1C–D*). In ICD layer, the N-terminal ankyrin repeats domain (ARD) is below the linker-helix domain (LHD) region (*Figure 1E*). The TRP helix mediates interactions with TMD. The C-terminal helix 1 (CH1) and the C-terminal helix 2 (CH2) fold back to interact with LHD and ARD (*Figure 1E*). In TMD layer, the pre-S1 elbow, the voltage sensor-like domain (VSLD, S1-S4), and the pore domain (S5-S6) assemble in a domain-swapped fashion. Densities of several putative

lipid molecules were readily observed in the TMD layer, most of which are in the grooves between VSLD and the pore domains (*Figure 1C–D*).

## The CMZ-binding site

By comparing the cryo-EM maps of hTRPC5 in complex with CMZ with both the apo state and HC-070-bound state, we found a nonprotein density within a pocket surrounded by S1-S4 and TRP re-entrant loop of VSLD in the CMZ-bound state (*Figure 2A–B*, *Figure 1—figure supplement 5E–H*). Moreover, the size and shape of this density match CMZ, suggesting this density represents CMZ. Several residues in hTRPC5 are close to CMZ molecule (*Figure 2C*), including Y374 on S1, F414, G417 and E418 on S2, D439, M442, N443 and Y446 on S3, R492, S495 and L496 on S4, and P659 on TRP re-entrant loop (*Figure 2C–D*). Among them, F414 forms a π-π stacking interaction with the chlorophenyl ring of CMZ. Side chains of Y374, M442, Y446, and P659 show hydrophobic interactions with CMZ. Residues surrounding the CMZ-binding site are highly conserved between TRPC4

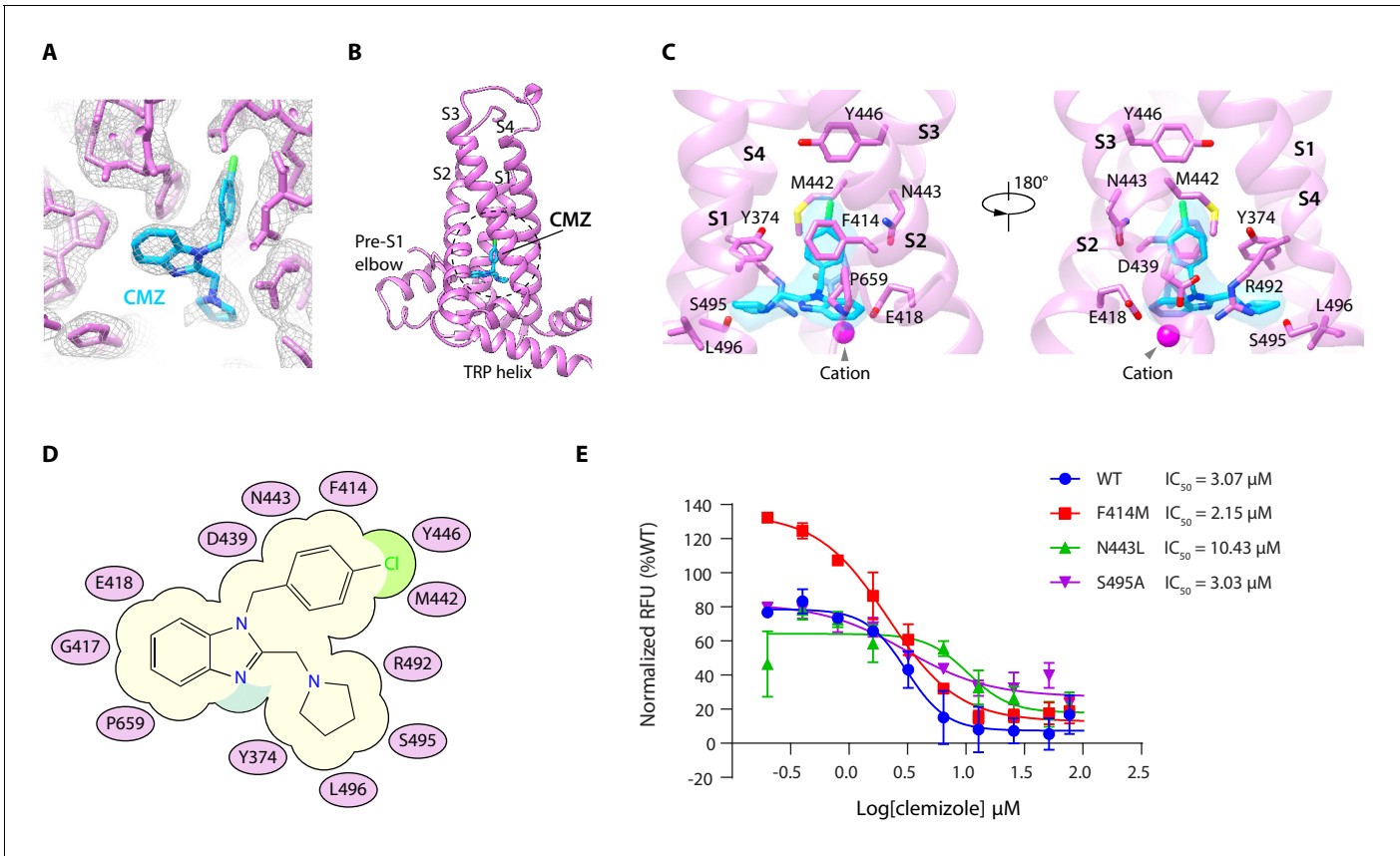

**Figure 2.** Clemizole (CMZ) binding site in human TRPC5 (hTRPC5). (**A**) Densities of CMZ and nearby residues. The map is contoured at 5.0 σ (gray mesh). CMZ and hTRPC5 are shown as sticks, colored in sky blue and purple, respectively. (**B**) Overview of the CMZ-binding site in hTRPC5. The dashed region denotes the binding pocket of CMZ. (**C**) Close-up view of the CMZ-binding site. CMZ is shown as transparent surface superimposed with sticks. Side chains of residues that interact with CMZ are shown as sticks. A cation near CMZ is shown as a magenta sphere. (**D**) Cartoon representation of the interactions between CMZ and hTRPC5. S1-S4 of subunit A are represented as purple ovals. Residues that interact with CMZ are labeled inside the ovals. (**E**) Inhibitory effect of CMZ on various hTRPC5 mutants, measured by FLIPR calcium assay (data are shown as means ± standard error, n = 3 independent experiments).

The online version of this article includes the following source data and figure supplement(s) for figure 2:

**Source data 1.** Inhibitory effect of CMZ on various hTRPC5 mutants.
**Figure supplement 1.** Sequence alignments of the transient receptor potential canonical (TRPC) channels.
**Figure supplement 2.** Functional characterization of human TRPC5 (hTRPC5) mutants.
**Figure supplement 2—source data 1.** Activation of hTRPC5 mutants by 14 mM $Ca^{2+}$ normalized to wild type hTRPC5.
**Figure supplement 2—source data 2.** Activation of hTRPC5 mutants by extracellular $Ca^{2+}$ normalized to wild type hTRPC5.

and TRPC5, in agreement with their similar selectivity for CMZ type inhibitors (*Figure 2—figure supplement 1*). To understand the function of CMZ interacting residues and the TRPC subtype selectivity of CMZ, we mutated CMZ interacting residues into alanines or the counterparts of TRPC3/6/7. Although all of the mutants showed surface expression (*Figure 2—figure supplement 2F–G*), some of them had much lower response to 14 mM extracellular calcium compared to WT channel (*Figure 2—figure supplement 2A*). We only analyze mutants which showed more than half of WT's response to 14 mM extracellular $Ca^{2+}$. We found mutations of F414M and S495A did not change the potency of CMZ much (*Figure 2E*, *Table 2*), while N443L decrease the potency (*Figure 2E*, *Table 2*), suggesting N443 is one of the structural determinants of TRPC subtype selectivity for CMZ inhibition. Although N443 is not involved in polar interactions with CMZ, but it is in close proximity to CMZ and probably has Van der Waals interactions with CMZ. Replacement of N443 with Leu might alter these interactions. These results collectively confirmed this is the CMZ-binding site which carries the inhibitory function against hTRPC5.

## The HC-070-binding site

A nonprotein density surrounded by the S5, pore helix from one subunit, and S6 of the adjacent subunit was observed in the presence of HC-070, compared with the apo state and CMZ-bound state (*Figure 3A–B*, and *Figure 1—figure supplement 7G–I*). Furthermore, the size and shape of this density perfectly match those of HC-070, indicating this density represents HC-070 compound. HC-070 is a clover-shaped molecule with a hydroxypropyl tail, a chlorophenyl ring, and a chlorophenoxy ring, all of which extend out from the central methylxanthine core. hTRPC5 makes extensive interactions with HC-070 (*Figure 3C–D*). The side chain of F576 on pore helix forms π-π stacking interaction with the methylxanthine core of HC-070. R557 on the loop linking S5 and pore helix (linker), Q573 and W577 on pore helix form polar contacts with the hydroxypropyl tail of HC-070. F599, A602, and T603 on S6 from the adjacent subunit also stabilize the hydroxypropyl tail. F569 and L572 interact hydrophobically with the chlorophenyl ring of HC-070 (*Figure 3C and D*). The chlorophenoxy branch is close to C525, which might account for its weaker density and higher flexibility compared to other part of HC-070 (*Figure 1—figure supplement 7F*). Residues involved in HC-070 binding are absolutely conserved between hTRPC4 and hTRPC5 but not in TRPC3/6/7 (*Figure 2—figure supplement 1*), in agreement with the fact that HC-070 selectively inhibits TRPC4/5 with high potency compared with TRPC3/6/7 (*Just et al., 2018*). To understand the contribution of hTRPC5 residues on the inhibitory effect of HC-070, we mutated HC-070 interacting residues into alanines and only analyze mutants with more than 50% of WT's response to 14 mM extracellular $Ca^{2+}$ (*Figure 2—figure supplement 2A*). We found that C525A does not change the potency of HC-070 much (*Figure 3E*, *Table 3*), while R557A, F569A, L572A, Q573A, F599A, and T603A show decreased potency (*Figure 3E–F*, *Table 3*). Notably, Q573A dramatically decreased HC-070 potency (*Figure 3F*, *Table 3*), emphasizing the essential role of the hydrogen bond between Q573 and hydroxypropyl group of HC-070. We also mutated F599 of hTRPC5 into the Asn (N in TRPC3/6) and found that F599N increased the $IC_{50}$ more dramatically than F599A did, probably because F599N mutation perturbs the local hydrophobic environment of the HC-070-binding site and indirectly affects ligand binding (*Figure 3F*).

**Table 2.** The potency of clemizole (CMZ) on various human TRPC5 (hTRPC5) constructs.

| TRPC5 constructs | CMZ | | |
| --- | --- | --- | --- |
|  | LogIC$_{50}^{*}$ | Hill slope | IC$_{50}$ (μM) |
| hTRPC5 WT | 0.49 ± 0.08 | −2.52 | 3.07 |
| Mutations in CMZ-binding pocket | | | |
| F414M | 0.33 ± 0.06 | −1.45 | 2.15 |
| N443L | 1.02 ± 0.15 | −2.42 | 10.43 |
| S495A | 0.48 ± 0.15 | −1.21 | 3.03 |

* Data were expressed as logIC$_{50}$ ± SEM, n = 3.

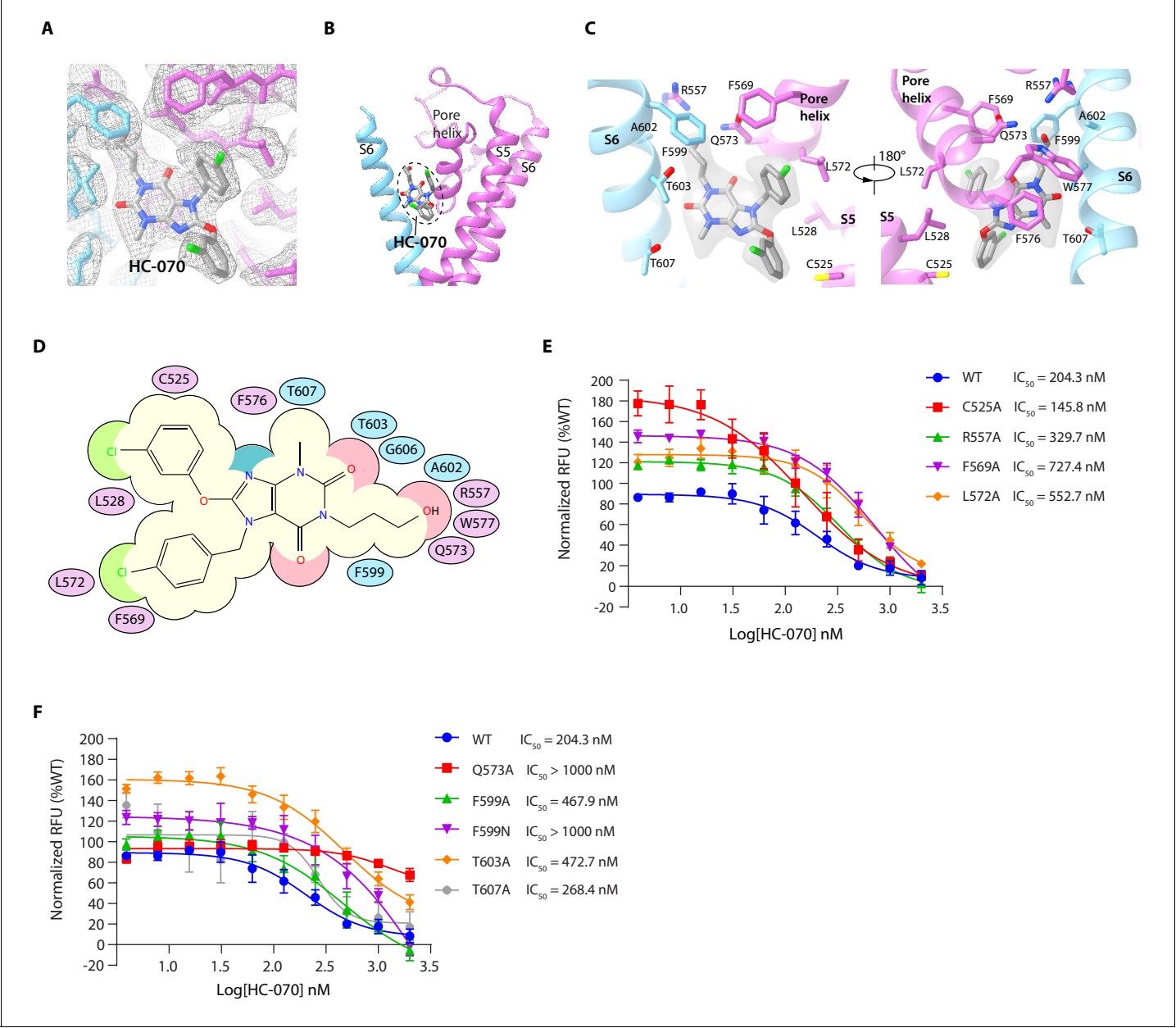

**Figure 3.** HC-070 binding site in human TRPC5 (hTRPC5). (**A**) Densities within the HC-070-binding site. The map is contoured at 3 σ (gray mesh). HC-070 and hTRPC5 are shown in sticks, with HC-070 colored in gray. Subunits A and B are colored in purple and light blue, respectively. (**B**) Overview of the HC-070-binding site in hTRPC5. The dashed region marks the binding pocket of HC-070. (**C**) Close-up view of the HC-070-binding site. HC-070 is shown as transparent surface superimposed with stick model. Side chains of residues that interact with HC-070 are shown as sticks. (**D**) Cartoon representation of the interactions between HC-070 and hTRPC5. S5 of subunit A and S6 of subunit B are represented as purple and light blue ovals, respectively. Residues that interact with HC-070 are labeled inside the ovals. (**E, F**) Inhibitory effect of HC-070 on various hTRPC5 mutants, measured by FLIPR calcium assay (data are shown as means ± standard error, n = 3 independent experiments).

The online version of this article includes the following source data for figure 3:

**Source data 1.** Inhibitory effect of HC-070 on various hTRPC5 mutants.

## Ion conduction pore and cation-binding sites

In TMD, pore helices, pore loops, and S6 of four protomers form the transmembrane pathway for ion permeation. Calculated pore profiles showed that the constriction is formed by the side chains of I621, N625, and Q629, which represents the lower gate of hTRPC5 (*Figure 4A–B*). The smallest radius at this gate is below 1 Å, indicating the channel is in a nonconductive closed state

**Table 3.** The potency of HC-070 on various human TRPC5 (hTRPC5) constructs.

| TRPC5 constructs | HC-070 | | |
| --- | --- | --- | --- |
| | LogIC$_{50}$* | Hill slope | IC$_{50}$ (nM) |
| hTRPC5 WT | 2.31 ± 0.10 | −1.52 | 204.3 |
| Mutations in HC-070-binding pocket | | | |
| C525A | 2.16 ± 0.15 | −1.03 | 145.8 |
| R557A | 2.52 ± 0.07 | −1.45 | 329.7 |
| F569A | 2.86 ± 0.15 | −1.21 | 727.4 |
| L572A | 2.74 ± 0.17 | −1.58 | 552.7 |
| Q573A | 3.10 ± 0.61 | −1.75 | >1000 |
| F599A | 2.67 ± 0.29 | −1.09 | 467.9 |
| F599N | 3.57 ± 1.55 | −0.86 | >1000 |
| T603A | 2.68 ± 0.16 | −1.20 | 472.7 |
| T607A | 2.43 ± 0.12 | −3.13 | 268.4 |

* Data were expressed as logIC$_{50}$ ± SEM, n = 3.

(*Figure 4B*). This is in agreement with the inhibitory function of CMZ and HC-070. A putative cation density inside VSLD, surrounded by E418 and E421 on S2 and N436, D439 on S3, was observed in both CMZ and HC-070 maps (*Figure 4C*). Strikingly, this putative cation is in close proximity to CMZ in CMZ-bound hTRPC5 (*Figure 2C*). Similar cation density was previously observed in mTRPC4 and mTRPC5 and identified as activating calcium in TRPM2, TRPM4, and TRPM8 (*Autzen et al., 2018*; *Duan et al., 2019*; *Duan et al., 2018*; *Huang et al., 2018*; *Yin et al., 2019*). We generate the E418Q-E421Q-D439N triple mutant (*Duan et al., 2019*) to abolish the cation-binding capability of this site and found the channel cannot be activated by extracellular calcium anymore (*Figure 2—figure supplement 2E*, *Figure 4—figure supplement 1B*), suggesting the essential role of this site for calcium activation. Since icilin activates TRPM8 in a Ca$^{2+}$-dependent manner and binds to a topologically equivalent site as CMZ (*Yin et al., 2019*), we carried out electrophysiology experiment and used 1 mM EGTA and 1 mM EDTA to chelate intracellular Ca$^{2+}$ to explore whether Ca$^{2+}$ is required for CMZ inhibition of TRPC5. We found that CMZ could inhibit EA-elicited currents in this condition (*Figure 4—figure supplement 1C*), suggesting CMZ exerts its function independent of high intracellular Ca$^{2+}$.

In ICD, there is another extra density found in the ankyrin repeat 4 (AR4)-linker helix (LH1) region (*Figure 4D*). The ion is coordinated by the side chains of H178 and C182 on the AR4-LH1 loop and C184 and C187 on the LH1 helix (*Figure 4D*). Based on the chemical environments for ion coordination, this is likely to be an HC3-type zinc-binding site, as observed previously in a bacteria glucokinase (*Miyazono et al., 2012*). Because the zinc ions were from an endogenous source and copurified with hTRPC5, we speculate that this is a high affinity zinc- binding site that constitutively chelates zinc. The residues coordinating the putative zinc ion are highly conserved in TRPC ion channels (*Figure 2—figure supplement 1*), and mutation of C176A-C178A-C181A does not affect calcium activation or EA activation (*Figure 2—figure supplement 2E*, *Figure 4—figure supplement 1A–B*), suggesting that this zinc-binding site might play other regulatory or structural roles, instead of ion channel gating.

## The lipid-binding sites

In the 2.7 Å cryo-EM maps of hTRPC5 in complex with CMZ, we observed several putative lipid molecules in the TMD (*Figure 1C–D*, *Figure 4E*). One lipid is located near the S5 and pore helix from one subunit and S6 from the adjacent subunit in the CMZ-bound hTRPC5. Its density highly resembles diacylglycerol (DAG) molecule which shows two acryl tails, but without phosphate group or headgroup of typical phospholipids (*Figure 4F*, *Figure 4—figure supplement 2A*). In agreement with our structural observation, further HPLC-MS experiments revealed that the purified protein sample contained markedly higher level of DAG in comparison with the purification buffer control (*Figure 4—figure supplement 3*). Therefore, we tentatively assign this density as the DAG. The free

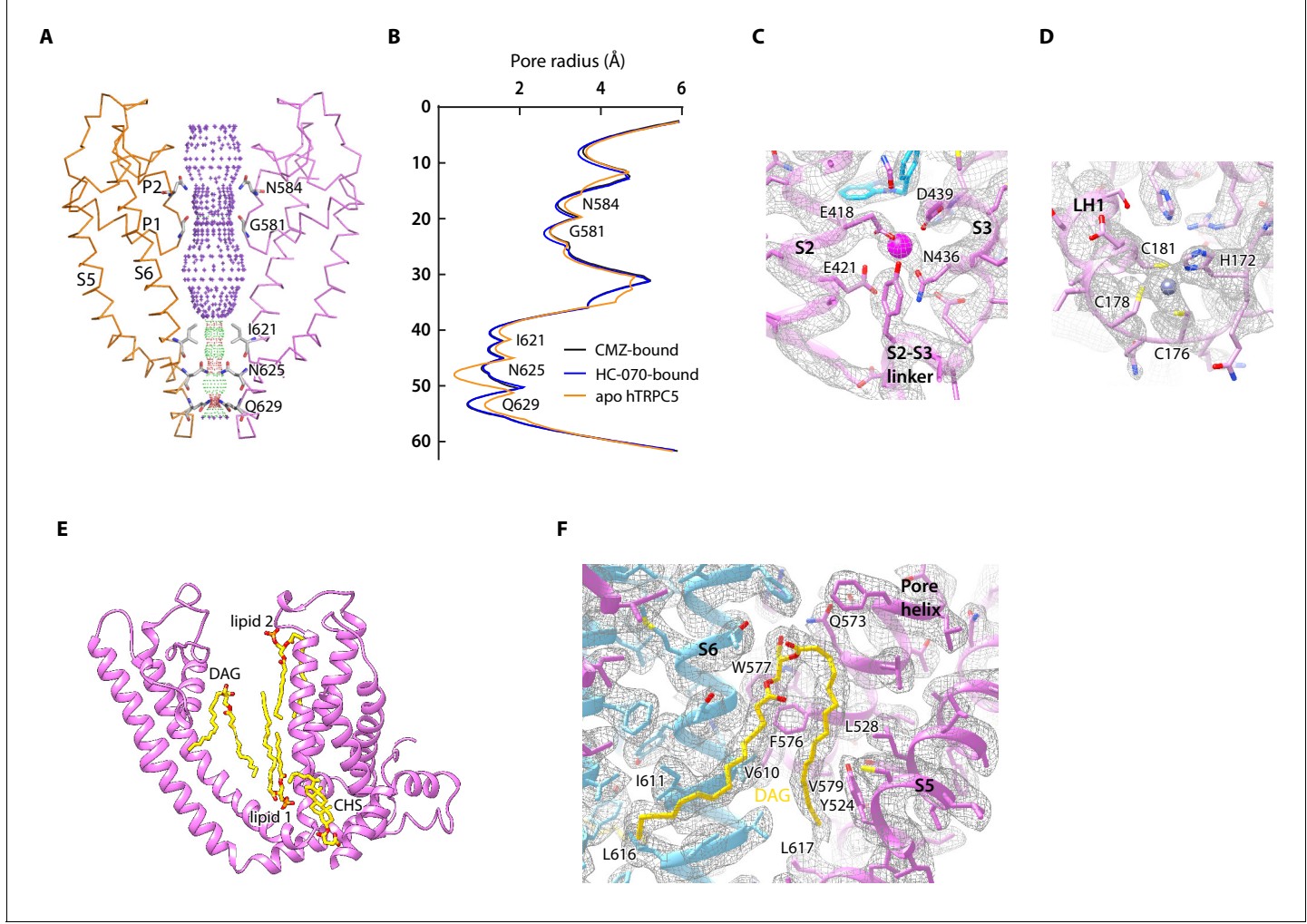

**Figure 4.** Ion conduction pore and other nonprotein densities in human TRPC5 (hTRPC5) maps. (A) Side view of the pore region of nonconductive hTRPC5 (clemizole [CMZ]-bound). Subunits A and C are shown as ribbon and colored the same as in *Figure 1*. Subunits B and D are omitted for clarity. Ion conduction pathway along the pore is shown as dots with key residues labeled, calculated by HOLE. Purple, green, and red dots define pore radii of >2.8, 1.4–2.8, and <1.4 Å, respectively. (B) Calculated pore radii of CMZ-bound hTRPC5, HC070-bound hTRPC5, and apo hTRPC5 are shown vertically. (C) Close-up view of the putative cation-binding site in transmembrane domain (TMD). Densities of the cation and the related residues are contoured at 4.6 σ and shown as gray mesh. The side chains of the residues that interact with the cation are shown as sticks. (D) Close-up view of the putative zinc-binding site in hTRPC5. Densities of the zinc ion and interacting residues are contoured at 4.3 σ and shown as gray mesh. (E) Other modeled nonprotein densities in TMD in CMZ-bound hTRPC5 are shown as sticks and colored the same as in *Figure 1*. (F) Close-up view of the putative diacylglycerol (DAG)-binding sites in hTRPC5. Densities of the DAG and interacting residues are contoured at 5 σ and shown as gray mesh. Residues that interact with DAG were labeled.

The online version of this article includes the following figure supplement(s) for figure 4:

**Figure supplement 1.** Functional characterization of various human TRPC5 (hTRPC5) mutants related to ion-binding sites.

**Figure supplement 2.** Lipid densities in human TRPC5 (hTRPC5) map.

**Figure supplement 3.** The identification of diacylglycerol (DAG) molecule in protein sample.

hydroxyl group of this DAG molecule forms polar contacts with Q573 (*Figure 4F*). Its two hydrophobic tails interact with several hydrophobic residues: L528, Y524 on S5; F576, W577, and V579 on the pore helix; L617 on S6; and V610, I611, and L616 on S6 from adjacent subunit. We speculate that the DAG molecules observed here were carried over from host cells throughout purification procedure, because similar densities were also reported previously in the cryo-EM map of mTRPC4 and mTRPC5 (*Duan et al., 2019*; *Duan et al., 2018*). Unexpectedly, in the map of HC-070-bound hTRPC5, we found that the binding site of glycerol group of DAG was occupied by HC-070, while the binding site of two DAG tails was still filled by lipid-like densities (*Figure 1—figure supplement*

*7G*), indicating the binding of the glycerol group of DAG and HC-070 are mutually exclusive. We observed additional lipids densities in both CMZ-bound hTRPC5 and HC-070-bound hTRPC5 maps. Lipid 1 localizes in the inner leaflet and is bound by S3, S4, and S5 from one subunit and S5, S6 from adjacent subunit (*Figure 4E*, *Figure 4—figure supplement 2B*). Lipid 2 is in the outer leaflet of the membrane and bound by S3 and S4 (*Figure 4E*, *Figure 4—figure supplement 2C*). There is one putative cholesteryl hemisuccinate (CHS) molecule which is sandwiched by S1, S4 from one subunit and S5 from adjacent subunit (*Figure 4E*, *Figure 4—figure supplement 2D*). Densities similar to the putative CHS were observed previously in corresponding positions in TRPM4 and TRPCs (*Autzen et al., 2018*; *Duan et al., 2019*; *Duan et al., 2018*; *Tang et al., 2018*). It is previously reported that this putative CHS-binding site is important for the activation of mTRPC5 (*Duan et al., 2019*).

## Structure comparison of hTRPC5 structures at different states

The overall structures of hTRPC5 at different states are highly similar, with RMSD of 1.456 Å between CMZ-bound hTRPC5 and apo state; 0.926 Å between HC-070-bound hTRPC5 and apo state, although there is a little structural variation in details (*Figure 5*). In the TMD, the bottom of S3 segment and the S2-S3 linker in HC-070-bound structure have small conformational change compared to the apo hTRPC5 (*Figure 5B*), which may be due to the HC-070 binding.

## Discussion

TRPC channels, especially TRPC5 and TRPC6, are putative drug targets. Recently, two distinct inhibitor-binding sites were identified in the hTRPC6 channel (*Bai et al., 2020*; *Tang et al., 2018*). During the editorial process of this work, inhibitor binding sites were also identified in TRPC4 (*Vinayagam et al., 2020*) and TRPC5 (*Wright et al., 2020*). Together with the sites identified in this study, we classify them into three groups, namely inhibitor-binding pocket (IBP) A-C (*Figure 6*, *Figure 6—figure supplement 1*). IBP-A inside the VSLD is accessible from the intracellular side (*Figure 6A*). Inhibitors that bind IBP-A, such as CMZ and GFB-8438, need to penetrate the plasma membrane first before reaching their binding site in VSLD. CMZ, M084, and AC1903 are

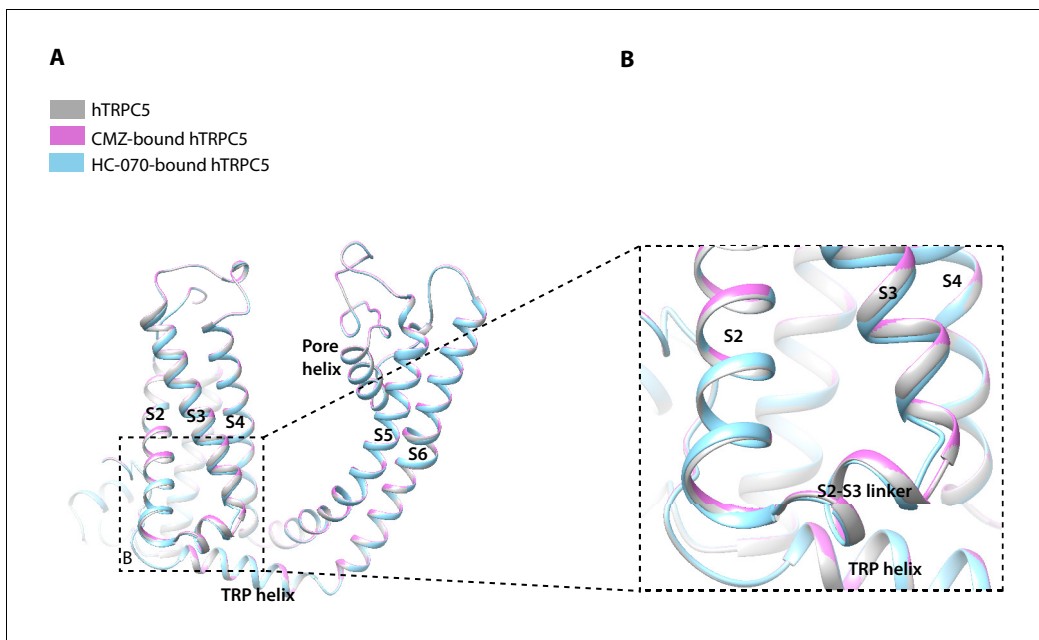

**Figure 5.** Structure comparison between clemizole (CMZ)-bound human TRPC5 (hTRPC5), HC-070-bound hTRPC5, and apo hTRPC5 in transmembrane domain (TMD). (**A**) The overall conformational difference in TMD between these three structures. All the structures are shown as cartoon, with apo hTRPC5 colored in gray, CMZ-bound hTRPC5 in purple, and HC-070-bound hTRPC5 in light blue. (**B**) Close-up view of the structure variations of bottom of S3 segment and the S2-S3 linker between HC-070-bound and apo hTRPC5 structures.

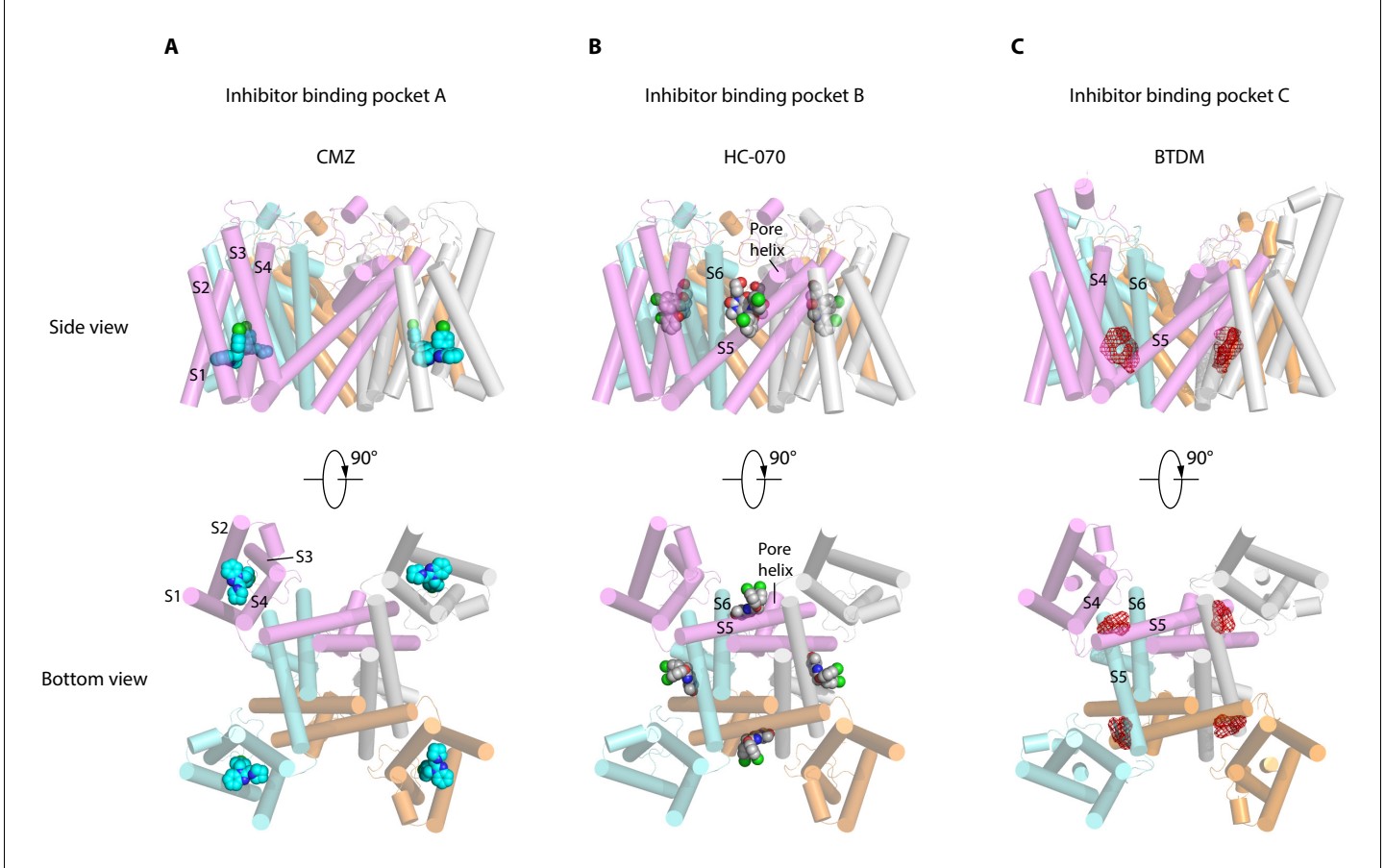

**Figure 6.** Inhibitor-binding pockets of transient receptor potential canonical (TRPCs) in transmembrane domain (TMD). (A) Clemizole (CMZ)-binding pocket in human TRPC5 (hTRPC5) is shown in side view and bottom view. The TRPC5 structure is shown as cylinder with CMZ compound shown as spheres and colored in sky blue. Each subunit of TRPC5 is colored the same as in *Figure 1*. (B–C) HC-070-binding pocket in hTRPC5 structure and BTDM-binding pocket in TRPC6 structure are shown in side view and bottom view. HC-070 is shown as spheres and colored in gray. The densities of BTDM are shown as red mesh.

The online version of this article includes the following figure supplement(s) for figure 6:

**Figure supplement 1.** Ligand-binding sites in transient receptor potential (TRP) family.

**Figure supplement 2.** Comparison of chemical structures of different TRPC5 ligands.

benzimidazole derivatives which can inhibit TRPC5 with $IC_{50}$ values at 1.0–1.3, 8.2, and 14.7 µM, respectively (*Richter et al., 2014*; *Zhou et al., 2017*; *Zhu et al., 2015*). Their similar chemical structures suggest that they probably share a common binding site at IBP-A (*Figure 6—figure supplement 2A–C*). CMZ is 10- and 26-fold more potent against TRPC5 than TRPC3/6 and TRPC7, respectively (*Richter et al., 2014*). Among the CMZ interacting residues, E418, D439, M442, R492 are identical between TRPC4/5 and TRPC3/6/7, whereas residue Y374 in TRPC5 is Phe, F414 is Met, N443 is Leu, Y446 is Phe, S495 is Tyr, and L496 is Met in TRPC3/6/7 (*Figure 2—figure supplement 1*). We found that F414M mutation in hTRPC5 does not change the potency of CMZ, while N443L decreases its potency (*Figure 2E*, *Table 2*), highlighting the importance of N443 on CMZ selectivity. GFB-8438, GFB-9289, and GFB-8749 are piperazinone/pyridazinone derivatives that inhibit TRPC4 and TRPC5 channels. Structure of zebrafish TRPC4 in complex with these inhibitors showed that they also bind inside VSLD (*Vinayagam et al., 2020*). GFB-8438 interacts with H369, Y646, L495, Y373, S488, M441, N442, F413, and R491 of zebrafish TRPC4 which are equivalent to H370, Y650, L496, Y374, S489, M442, N443, F414, and R492 in hTRPC5 (*Vinayagam et al., 2020*). Therefore, both CMZ and GFB-8438 share a common set of interacting residues on hTRPC5 including L496, Y374, M442, N443, and R492. More importantly, both CMZ and GFB-8438 have several nonoverlapping

interacting residues on hTRPC5, suggesting that structure-based hybridization of CMZ and GFB-8438 might exploit more interacting residues on hTRPC5 and thus enhance the inhibitor potency and selectivity. Although VSLD has lost the voltage sensing function, it plays an important structural and functional role in TRP channels. Based on the gating model of TRPV1, channel activation is associated with relative movements between VSLD and pore (*Cao et al., 2013*; *Gao et al., 2016*) and thus, stabilization of VSLD in an inactive conformation might be a common scheme to inhibit channel opening. Indeed, AM-1473, a potent antagonist of hTRPC6, was found to bind at IBP-A in hTRPC6 as well (*Bai et al., 2020*). Moreover, IBP-A is the AMTB and TC-I 2014-binding sites in TRPM8 (*Diver et al., 2019*), 2-APB-binding site in TRPV6 (*Singh et al., 2018*; *Figure 6—figure supplement 1A*). The studies suggest that VSLD of TRP channels is a hotspot for drug discovery.

IBP-B located in the pore domain is sandwiched at the subunit interface (*Figure 6B*). HC-070 binds at IBP-B which is close to the extracellular side. Strikingly, IBP-B is partially occupied by the glycerol group of DAG molecule in the map of CMZ-bound hTRPC5 or apo hTRPC5 (*Figure 1—figure supplement 7G–I*). Previous studies showed that some mutations in this lipid-binding site in mTRPC5, such as F576A and W577A double mutations, render mTRPC5 unresponsive to EA stimulation (*Duan et al., 2019*). This lipid-binding site also plays an important role in the gating of DAG-activated hTRPC3/6 channels. Mutation of W680 to alanine in hTRPC6, corresponding to W577A in hTRPC5, abolished OAG activation of hTRPC6 (*Bai et al., 2020*). Mutations of glycine residues in hTRPC6/3, corresponding to G606 in hTRPC5, blunted PLC-mediated activation of TRPC6/3, altered responses to DAGs of TRPC3, and also changed the pattern of photoactivation of TRPC6/3 by a photo-switchable DAG analogue, OptoDArG (*Lichtenegger et al., 2018*). Mutation of this glycine residue into alanine in TRPC3 removed channel desensitization upon PLC activation and increased efficacy of GSK1702934A by maintaining the long-open state (*Svobodova et al., 2019*). E672A, F675A, W680A, N702A, Y705A, and V706A mutants of TRPC6 (corresponding to F569, L572, W577, F599, A602, and T603 in hTRPC5) maintained normal membrane surface expression but completely abolished the response to OAG (*Bai et al., 2020*). These data concertedly suggest that the DAG-binding pocket observed here in hTRPC5 is also structurally conserved in TRPC3/6 and is essential for TRPC3/6 channel activation by DAG and its analogues, such as OAG. Pharmacological compounds binding at IBP-B might affect TRPC channel gating either by acting on TRPC directly or by perturbing the normal function of DAG. Here, we observed that the binding of HC-070 at this site replaces the glycerol group of DAG, stabilizes hTRPC5 in a closed state, and thus inhibits channel opening. The residues involved in HC-070 binding are absolutely identical between hTRPC5 and hTRPC4, but they share only 20% similarities between TRPC4/5 and TRPC3/6/7 (*Figure 2—figure supplement 1*), which explains the high selectivity of HC-070 for TRPC4/5 over TRPC3/6/7 (*Just et al., 2018*). In addition, HC-070 shows inhibitory effect on heterotetrameric hTRPC1: C5 and hTRPC1: C4 with relatively lower potency compared to homotetrameric hTRPC4/5 (*Just et al., 2018*), probably because of the lower sequence identity between TRPC4/5 and TRPC1 (46% according to the sequence alignment) (*Figure 2—figure supplement 1*). Indeed, through mutagenesis, we found that many residues participating in interactions between HC-070 and hTRPC5 are sensitive to alanine mutations (*Figure 3E–F*). It is reported that Pico145, an HC-070 analogue, is more potent against hTRPC4/5 than HC-070 (*Just et al., 2018*), possibly due to the substitution of chloride with trifluoromethoxy group on the chlorophenoxy ring of HC-070 (*Figure 6—figure supplement 2G–H*). Indeed, recent structure study showed that Pico145 binds to hTRPC5 channel using a similar pose as HC-070, except the chlorophenoxy ring with a trifluoromethoxy group on it inserts into a cavity surrounded by L521, Y524, F576, and V610 of hTRPC5 (*Figure 1—figure supplement 7J*; *Wright et al., 2020*). Another HC-070 analogue, AM237, is a TRPC5 activator with $EC_{50}$ around 15–20 nM (*Minard et al., 2019*), and AM237 activates homomeric hTRPC5 channel instead of the heterotetrameric TRPC1: C5 channel (*Minard et al., 2019*), suggesting that AM237 also binds at IBP-B which is at the interface between adjacent subunits. Compared with Pico145, AM237 has an additional chloride atom on the phenoxy ring (*Figure 6—figure supplement 2I*), which makes the phenoxy group larger and this chloride atom might be in clash with L521 of hTRPC5. Therefore, AM237 might activate hTRPC5 by pushing L521 on S5 using its chloride-phenoxy group (*Wright et al., 2020*). Consistent with our results of HC-070-binding site mutants, mutations in IBP-B, including Q573T, F576A, and W577A, decrease the potency of AM237 on hTRPC5 (*Wright et al., 2020*). Recently, it is reported that an hTRPC6 agonist AM-0883 binds at IBP-B as well (*Figure 6—figure supplement 1B*; *Bai et al., 2020*). The identification of DAG lipid, TRPC inhibitor, and TRPC

activator binding at IBP-B suggests that it is a common modulatory ligand-binding site in TRPC channels.

IBP-C is between VSLD and pore domain, formed by S3, S4, S4-S5 linker from one subunit and S5-S6 from the adjacent subunit (*Figure 6C*). BTDM, a high affinity inhibitor of hTRPC6, binds at IBP-C (*Figure 6C*, *Figure 6—figure supplement 1C*; *Tang et al., 2018*). When BTDM is not supplied, IBP-C is occupied by a putative phospholipid (*Bai et al., 2020*). In TRPC4/5, mutations of G503/G504 to serine on S4-S5 linker lead to constitutive activation of the channels (*Beck et al., 2013*). In addition, vanilloid agonist resiniferatoxin (RTX), vanilloid antagonist capsazepine, and an endogenous phosphatidylinositol bind at IBP-C on TRPV1 (*Figure 6—figure supplement 1C*; *Cao et al., 2013*; *Gao et al., 2016*). These data together suggest that IBP-C is important for the gating of both TRPC and TRPV channels.

In summary, our structures of hTRPC5 reported here have uncovered the binding pockets for two distinct classes of inhibitors in high-resolution details. Moreover, the structures suggest that these inhibitors inhibit the hTRPC5 channel by stabilizing it in an apo-like closed state. Further structural comparisons with previously reported TRPC channel structures have revealed three common IBPs in TRPC channels. Our studies paved the way for further mechanistic investigations and structure-based drug development.

# Materials and methods

## Key resources table

| Reagent type (species) or resource | Designation | Source or reference | Identifiers | Additional information |
|---|---|---|---|---|
| Gene (*Homo sapiens*) | hTRPC5 | This paper | NCBI Reference sequence: NM_012471 | Provided by Dr. Xiaolin Zhang, Dizal Pharmaceutical Company |
| Recombinant DNA reagent | hTRPC5 | This paper | | Subcloned into a pBM Bacmam vector with N-terminal GFP-MBP tag |
| Recombinant DNA reagent | hTRPC5 | This paper | | Subcloned into a pCDNA3.1 vector with no tag |
| Recombinant DNA reagent | hTRPC5$_{1-764}$ | This paper | | Subcloned into a pBM Bacmam vector with N-terminal GFP-MBP tag |
| Cell line (FreeStyle 293 F) | FreeStyle 293 F | Thermo Fisher Scientific | R79007 | RRID:CVCL_D603 |
| Cell line (AD-293) | AD-293 | Agilent | 240085 | RRID:CVCL_9804 |
| Cell line (Sf9) | Sf9 | Thermo Fisher Scientific | 12659017 | RRID:CVCL_0549 |
| Antibody | Anti-TRPC5 (rabbit polyclonal) | Proteintech | 25890–1-AP | 1:1000 RRID:AB_2880285 |
| Antibody | Anti-N-Cadherin (rabbit polyclonal) | Proteintech | 22018–1-AP | 1:3000 RRID:AB_2813891 |
| Chemical compound, drug | Clemizole | CAYMAN | 17695 | |
| Chemical compound, drug | HC-070 | MedChemExpress (MCE) | 31244 | |
| Chemical compound, drug | (-)-Englerin A | PanReac AppliChem | A8907 | |
| Software, algorithm | Gctf_v1.18 | *Zhang, 2016* PMID:26592709 | | |
| Software, algorithm | Gautomatch v0.56 | K. Zhang, MRC LMB (https://www2.mrc-lmb.cam.ac.uk/research/locally-developed-software/zhang-software/) | | |
| Software, algorithm | RELION-3.0 | *Zivanov et al., 2018* PMID:30412051 | | |

*Continued on next page*

*Continued*

| Reagent type (species) or resource | Designation | Source or reference | Identifiers | Additional information |
|---|---|---|---|---|
| Software, algorithm | PHENIX 1.18.1–3865 | *Adams et al., 2010*<br>PMID:20124702 | | |
| Software, algorithm | HOLE2 v2.2.005 | *Smart et al., 1996*<br>PMID:9195488 | | |
| Software, algorithm | cryoSPARC v3.1 | *Punjani et al., 2017*<br>PMID:28165473 | | |
| Software, algorithm | Chimera-1.13 | *Pettersen et al., 2004*<br>PMID:15264254 | | |
| Other | Rhod-2/AM | AAT Bioquest | 21062 | |

## Cell culture

Sf9 cells (from Thermo Fisher Scientific) were cultured in Sf-900 III SFM medium (Gibco) at 27°C. HEK 293 F cells (from Thermo Fisher Scientific) grown in Free Style 293 medium + 1% FBS at 37°C were transfected for electrophysiology and FSEC. HEK 293 F cells grown in SMM-293TI medium + 1% FBS at 37°C were used for baculovirus infection and protein purification. AD293 cells (from Agilent) were cultured in basic DMEM (Gibco) + 10% FBS medium at 37°C for FLIPR calcium assay and surface expression level detection. All these cell lines were free of mycoplasma contamination detected by MycoBlue Mycoplasma Detector assay (Vazyme, D101-02), but their identities were not further authenticated.

## FSEC

The cDNA of hTRPC5 and related mutants were cloned into a homemade BacMam vector with N-terminal GFP-MBP tag (*Li et al., 2017*). The expression levels of various hTRPC5 constructs were screened by FSEC (*Kawate and Gouaux, 2006*). Expression constructs were transfected into HEK293F cells using PEI. 40 hr post-transfection, cells were harvested by centrifuge at 5000 rpm for 5 min. Then the cells were solubilized by 1% (w/v) LMNG, 0.1% (w/v) CHS in TBS buffer (20 mM Tris-HCl, pH 8.0 at 4°C, and 150 mM NaCl) with 1 μg/ml aprotinin, 1 μg/ml leupeptin, 1 μg/ml pepstatin for 30 min at 4°C. Supernatants were collected after centrifuge at 40,000 rpm for 30 min and were loaded onto Superose-6 Increase (GE Healthcare) for FSEC analysis.

## Whole cell recording

To detect the inhibition of CMZ on TRPC5, we used EA to activate hTRPC5 in whole cell recordings; 800 ng of nontagged full-length hTRPC5 in pCDNA3.1 vector and 200 ng of home-made GFP-MBP vector were co-transfected into HEK293F cell. 40 hr post-transfection, the cells were plated on cover glass for recording. For the whole cell recording, the bath solution contains 140 NaCl, 5 CsCl, 10 EGTA, 1 MgCl$_2$, 10 glucose, and 10 HEPES (pH = 7.4, adjusted by NaOH) in mM and the pipette solution was 110 CsMe, 25 CsCl, 2 MgCl$_2$, 1 EDTA, 1 EGTA, and 30 HEPES (pH = 7.4 adjusted by CsOH) in mM to keep low calcium condition for the recording; 200 nM EA in bath solution was used for channel activation and 50 μM CMZ was used for channel inhibition. Patch electrodes were pulled by a horizontal microelectrode puller (P-1000, Sutter Instrument Co) to a tip resistance of 2.0–4.0 MΩ for recording. The TRPC5 currents were recorded in gap-free mode at holding potential of −60 mV through an Axopatch 200B amplifier (Axon Instruments). Data was further analyzed by pCLAMP 10.4 software. We observed similar results for three cells and one representative trace was shown.

## FLIPR calcium assay

AD293 cells cultured on 10 cm dish (grown in Dulbecco's modified Eagle's medium + 10% FBS + 13.9 μg/ml streptomycin + 6.7 μg/ml penicillin, 37°C) at 90-100% confluence were digested with trypsin-EDTA (0.25%) (Gibco) and suspended with 5 ml culture medium for each 10 cm dish. Then the suspended cell culture was further diluted for 5.4- to 7-fold with culture medium and seeded onto black well, clear bottom, poly-D-lysine-coated 96-well plate (Costar) and cultured overnight. We use full-length nontagged hTRPC5 construct for FLIPR assay unless indicated otherwise. hTRPC5

plasmids were transfected into AD293 cells (60-80% confluence) with PEI. 30–40 hr post-transfection, culture medium was discarded and cells were incubated with 50 µl per well of 4 µM Rhod-2/AM (AAT Bioquest) in TARODES's buffer (10 mM HEPES, 150 mM NaCl, 4 mM KCl, 2 mM $MgCl_2$, 2 mM $CaCl_2$, 11 mM glucose, pH 7.4) supplemented with 0.02% Pluronic F-127 (Sigma) for 5 min in dark at room temperature. Then dye was removed and replaced with 50 µl of TARODE's buffer per well. The application of compounds and detection of intracellular calcium elevation were performed using FLIPR-TETRA (Molecular Devices) at room temperature (25–27°C). Fluorescence signals were read with excitation/emission at 510–545/565–625 nm.

For dose-response activation curves of WT and mutants by extracellular $Ca^{2+}$, baseline fluorescence of the plate was measured for 30 s, followed by addition of 50 µl of TARODE's buffer containing a series of concentration of $Ca^{2+}$. Final concentrations for extracellular $Ca^{2+}$ were 2, 6.148, 8.222, 11.333, 16, 23, 33.5, and 49.25 mM. The normalized signal was calculated as follows: For each well, the average value of fluorescence from 0 to 20 s was considered as the 'start fluorescence'; the average of fluorescence from 309 to 328 s was considered as the 'end fluorescence'. The value of 'end fluorescence' minus 'start fluorescence' was considered as 'signal value' for each well. The 'signal value' of each construct activated by each concentration of $Ca^{2+}$ was divided by that of WT activated by 49.25 mM $Ca^{2+}$, resulted in '% WT'.

For the inhibition curves, baseline fluorescence of the plate was measured for 30 s, followed by addition of 50 µl of buffer A (TARODE's buffer containing a series of concentration of CMZ or HC-070). Fluorescence was measured for another 210 s, followed by addition of 50 µl of buffer B (TARODE's buffer containing additional high calcium and a series of concentration of CMZ or HC-070). Then fluorescence was measured for 300 s.

Buffer A contains 2× concentration of working concentration of inhibitor. As for CMZ, it was serially diluted to concentrations of: 205, 102.5, 51.25, 25.625, 12.8, 6.4, 3.2, 1.6, 0.8, 0.4 µM. As for HC-070, it was serially diluted to concentrations of: 4000, 2000, 1000, 500, 250, 125, 62.5, 31.25, 15.625, 7.81 nM. Buffer B is TARODE's buffer containing additional 36 mM $Ca^{2+}$ and a series of working concentration of inhibitor (102.5, 51.25, 25.625, 12.8, 6.4, 3.2, 1.6, 0.8, 0.4, 0.2 µM for CMZ and 2000, 1000, 500, 250, 125, 62.5, 31.25, 15.625, 7.81, 3.9 nM for HC-070, respectively). Final concentration of calcium for activation was 14 mM. The concentration series of CMZ was 102.5, 51.25, 25.625, 12.8, 6.4, 3.2, 1.6, 0.8, 0.4, 0.2 µM. The concentration series of HC-070 was 2000, 1000, 500, 250, 125, 62.5, 31.25, 15.625, 7.81, 3.9 nM. For each construct, there were two controls: for control-Max, buffer A was replaced by TARODE's buffer; for control-Min, both buffer A and buffer B were replaced by TARODE's buffer.

To calculate the normalized signal values for inhibition experiments, the averaged fluorescence value from 210 to 230 s of each well was considered as the 'start fluorescence'; the average of fluorescence value of last 20 time points was considered as the 'end fluorescence'. The value of 'end fluorescence' minus 'start fluorescence' was considered as 'signal value' for each well. The 'signal value' of each well was subtracted by 'signal value' of control-Min, resulting in a value considered as 'response value'. The 'response value' of each well was divided by 'response value' of control-Max to obtain an 'internal normalized response'. The 'internal normalized responses' were further multiplied by X% (X%: 'response value' of each construct activated by 14 mM $Ca^{2+}$ divided by that of WT, as shown in *Figure 2—figure supplement 2A*), resulting in a value considered as 'normalized RFU (% WT)'. $IC_{50}$ was obtained by fitting the data of '% WT' into function: $Y = Bottom + (Top - Bottom)/(1 + 10^{(LogIC50 - X) \cdot HillSlope})$.

Three independent experiments were performed on different days (biological replicates), with each experiment containing three replicates (technique replicates).

## Surface biotinylation and western blot

Thirty-six hours post-transfection, cells transfected with full-length nontagged TRPC5 plasmids in six-well plate were washed with PBS (pH = 8) for three times. Then 500 µl of 1.5 mM Sulfo-NHS-LC-biotin (APExBIO) in PBS (pH = 8) was added to each well and incubated at room temperature for 1 hr. Then the Sulfo-NHS-LC-biotin solution was discarded, followed by addition of ice-cold TBS +10 mM glycine (pH = 8) to terminate the reaction. Then the cells were washed with TBS and then lysed with TBS+1% MNG for 40 min on ice, followed by centrifugation at 16,125×$g$ for 20 min. Supernatants were incubated with streptavidin-agarose beads (SIGMA) for 2 hr at 4°C, followed by centrifugation at 1000×$g$ to remove flow-through. Beads were washed with lysis buffer for four times, followed by

elution with elution buffer (50 mM Tris pH 7.5, 150 mM NaCl, 3 mM biotin, 100 mM dithiothreitol [DTT], 2% SDS) at room temperature or 85°C for 7 min. The eluates were separated on 10% gradient SDS-PAGE gel and N-cadherin was detected by western blot using antibodies against N-cadherin (1:3000) (Proteintech) as an internal control to normalize loading volumes of the samples. Then TRPC5 protein was detected using antibodies against TRPC5 (1:1000) (Proteintech).

## Protein expression and purification

The BacMam expression system was used for large-scale expression of hTRPC5$_{1-764}$ as previously reported (*Tang et al., 2018*). Briefly, hTRPC5$_{1-764}$ was subcloned into a home-made BacMam vector with N-terminal GFP-MBP tag. The Bacmid was then generated by transforming this construct into DH10Bac *Escherichia coli* cells. Baculovirus was harvested about 7 days post-transfection of bacmid into Sf9 cells cultured in Sf-900 III SFM medium (Gibco) at 27 °C. P2 baculovirus was produced using Bac-to-Bac system. P2 virus was added to HEK293F cells at a ratio of 1:12.5 (v/v) when the cells were grown to a density of $2.0 \times 10^6$/ml in SMM 293T-I medium (Sino Biological Inc) supplemented with 1% FBS under 5% $CO_2$ in a shaker at 130 rpm at 37°C. After virus infection for 12 hr, 10 mM sodium butyrate was added and temperature was lowered to 30°C. Cells were harvested 48 hr post-infection and washed twice using TBS buffer. Cell pellets were collected and stored at −80°C for further purification.

Cell pellets were resuspended in TBS buffer supplemented with 1% (w/v) LMNG, 0.1% (w/v) CHS, 1 mM DTT, 1 mM phenylmethanesulfonylfluoride, and protease inhibitors, including 1 µg/ml aprotinin, 1 µg/ml leupeptin, 1 µg/ml pepstatin. The mixture was incubated at 4°C for 1 hr. After ultracentrifugation at 40,000 rpm for 1 hr in Ti45 rotor (Beckman), the supernatant was loaded onto 7 ml MBP resin. The resin was rinsed with wash buffer 1 (wash buffer 2 + 1 mM ATP + 10 mM MgCl$_2$) and wash buffer 2 (TBS + 40 µM glycol-diosgenin [GDN] + 0.005 mg/ml SPLE + 1 mM DTT) subsequently. Proteins were eluted with 80 mM maltose in wash buffer 2. The eluate was concentrated using a 100 kDa cut-off concentrator (Millipore) after digestion with H3C protease at 4°C overnight. hTRPC5 protein was further purified by size exclusion chromatography (Superose-6 Increase) in buffer containing TBS, 40 µM GDN, 0.005 mg/ml SPLE, and 1 mM tris(2-carboxyethyl) phosphine. The peak fractions corresponding to tetrameric TRPC5 channel were collected and concentrated to $A_{280} = 1.0$ with estimated concentration of 0.7 mg/ml for cryo-EM sample preparation.

## Cryo-EM sample preparation and data collection

Purified protein was first mixed with 100 µM $Ca^{2+}$ and then mixed with 500 µM CMZ or 50 µM HC-070, respectively. The sample with EA was prepared in two batches. In the first batch, protein added with 100 µM $Ca^{2+}$ was incubated with 6.5 µM EA for 30 min before concentration. In the second batch, 0.12 µl of 5 mM EA in 50% DMSO was added to 6 µl concentrated protein in the presence of 5 mM $Ca^{2+}$. We did not observe obvious difference in final cryo-EM reconstructions between the two EA samples. Therefore, we combined data collected from these two batches into the EA dataset. After incubation with ligands for 30 min on ice, the mixtures were ultracentrifugated at 25,000 rpm for 30 min in TLA55 rotor (Beckman) and the supernatants were used for cryo-EM sample preparation. Aliquots of 2.5 µl mixture were placed on graphene oxide grids as previously reported (*Phulera et al., 2018*). Grids were blotted at 100% humidity and flash-frozen in liquid ethane cooled by liquid nitrogen using Vitrobot Mark I (FEI). Grids were then transferred to a Titan Krios (FEI) electron microscope that was equipped with a Gatan GIF Quantum energy filter and operated at 300 kV accelerating voltage. Image stacks were recorded on a Gatan K2 Summit direct detector in super-resolution counting mode using SerialEM at a nominal magnification of 165,000× (HC-070-bound, calibrated pixel size of 0.821 Å/pixel) or 130,000× (CMZ-bound and apo state, calibrated pixel size of 1.045 Å/pixel), with a defocus ranging from −1.5 to −2.0 µm. Each stack of 32 frames was exposed for 7.1 s, with a total dose about 50 e$^-$/ Å$^2$ and a dose rate of 8 e$^-$ per pixel per second on detector.

## Image processing

All of the datasets were processed with similar workflow. The collected image stacks were motion-corrected by MotionCor2 (*Zheng et al., 2017*). After motion correction, good micrographs were manually selected, then GCTF (*Zhang, 2016*) was used for CTF estimation. Particles were auto-

picked using Gautomatch (kindly provided by Kai Zhang) based on the projection of hTRPC6 map (EMDB: EMD-6856), respectively. After one-round or two-rounds of reference-free two-dimensional (2D) classification, particles in classes with good features were selected for 3D classification with previous hTRPC6 map (EMDB: EMD-6856) low-pass filtered to 18 Å as the initial model using RELION-3.0 (*Zivanov et al., 2018*) with C1 symmetry. After 3D classification, particles in good classes with clearly visible α-helices in TMD were selected for further homogeneous refinements in cryoSPARC (*Punjani et al., 2017*) using C4 symmetry, because we found that refinement in cryoSPARC gave better Fourier shell correlations (FSCs) and maps for TRPC5 datasets. Reported resolutions are based on the gold-standard FSC 0.143 criterion after correction of masking effect (*Chen et al., 2013*). The map was sharpened with a B factor determined by cryoSPARC automatically.

### Model building, refinement, and validation

The previously reported TRPC4 (PDB: 5Z96) was used as the starting model and docked into the HC-070-bound or CMZ-bound hTRPC5 maps in Chimera (*Pettersen et al., 2004*), respectively. The fitted model was manually adjusted in COOT (*Emsley et al., 2010*), keeping the side chains of conserved residues and substituting nonconserved residues based on the sequence alignment between mTRPC5 and hTRPC5. The ligands models were generated using elbow module (*Moriarty et al., 2009*) in PHENIX (*Adams et al., 2010*). The ligands and phospholipids were manually docked into densities and refined using COOT. The models were further refined against the corresponding maps with PHENIX (*Adams et al., 2010*).

### Quantification and statistical analysis

The local resolution map was calculated using cryoSPARC. The pore radius was calculated using HOLE (*Smart et al., 1996*). The inhibition curves were calculated using GraphPad Prism 8. RMSD was calculated using PyMOL in all-atom mode.

### Identification of DAG molecule in hTRPC5 protein sample by LC-MS analysis

To prepare samples for LC–MS analysis, 20 μl of purified hTRPC5 (7 μM) in the wash buffer 1 (20 mM Tris-HCl, pH 8.0, 150 mM NaCl, 40 μM GDN, 0.005 mg/ml SPLE, and 1 mM DTT) was added into 400 μl isopropanol, vortexed and extracted by ultrasonication for 1 min. After centrifugation at 14,000 rpm for 10 min at 4°C, the supernatant was transferred into vial for further LC–MS analysis.

UPLC (Waters ACQUITY I-Class system) coupled with tandem ESI-Triple quadrupole mass spectrometry (Waters Xevo TQ-S Micro) was used to analyze 2 μl lipid extractants. An ACQUITY UPLC BEH Amide column (1.7 μm, 2.1 mm × 100 mm, Waters) was used on a HILIC mode for the separation of the lipid classes. The mobile phase of solvent A (10 mM ammonium acetate in 95:5 acetonitrile/water) and solvent B (10 mM ammonium acetate in 50:50 acetonitrile/water) was at a flow rate of 0.6 ml/min. The UPLC elution gradient was 0.1-20.0% B for 2 min, then 20-80% B for 3 min followed by 3 min re-equilibration. The eluted lipids were directly introduced into the mass spectrometer with a desolvation temperature of 500 °C and a capillary voltage of 2.8 kV under positive ESI mode, and the source temperature was set at 120°C. The analytes were monitored under full-scan mode with mass ranging from m/z 100 to 600, combined with a daughter scan from m/z 50 to 650 for fragmentor of 638.6(m/z), with a collision energy of 20 V. An MRM transition for DAG (18:1-18:1) quantitative detection was set as 638.6 to >339.3, with a core voltage of 50 V, and collision energy of 20 V Masslynx 4.0 was used to acquire and screen MS and MS/MS data. Notably, the available HPLC-MS results suggest markedly higher concentration of DAG molecule than purification buffer, but we cannot distinguish 1,2-DAG from its regioisomer 1,3-DAG with current available data. However, we found 1,2-DAG fits the electron density better, therefore, we modeled the electron density as 1,2-DAG molecule.

### Acknowledgements

The cDNAs of hTRPC5 were kindly provided by Dr. Xiaolin Zhang. We thank Dr. Yi Rao for sharing FLIPR instruments and Shangchen Han for technical supports. We thank the National Center for Protein Sciences at Peking University in Beijing, China, for assistance with negative stain EM. Cryo-EM data collection was supported by electron microscopy laboratory and Cryo-EM platform of Peking

University with the assistance of Xuemei Li, Daqi Yu, Xia Pei, Bo Shao, Guopeng Wang, and Zhenxi Guo. Part of structural computation was also performed on the Computing Platform of the Center for Life Science and High-performance Computing Platform of Peking University. This work is supported by grants from the Ministry of Science and Technology of China (National Key R and D Program of China, 2016YFA0502004 to LC), National Natural Science Foundation of China (91957201, 31870833, and 31821091 to LC, 31900859 to J-X W), and the China Postdoctoral Science Foundation (2016M600856, 2017T100014, 2019M650324, and 2019T120014 to J-XW). J-XW is supported by the Boya Postdoctoral Fellowship of Peking University and the postdoctoral foundation of the Peking-Tsinghua Center for Life Sciences, Peking University (CLS).

## Additional information

### Funding

| Funder | Grant reference number | Author |
|---|---|---|
| National Key Research and Development Program of China | 2016YFA0502004 | Lei Chen |
| National Natural Science Foundation of China | 91957201 | Lei Chen |
| National Natural Science Foundation of China | 31900859 | Jing-Xiang Wu |
| China Postdoctoral Science Foundation | 2016M600856 | Jing-Xiang Wu |
| China Postdoctoral Science Foundation | 2017T100014 | Jing-Xiang Wu |
| China Postdoctoral Science Foundation | 2019M650324 | Jing-Xiang Wu |
| China Postdoctoral Science Foundation | 2019T120014 | Jing-Xiang Wu |
| National Natural Science Foundation of China | 31870833 | Lei Chen |
| National Natural Science Foundation of China | 31821091 | Lei Chen |

The funders had no role in study design, data collection and interpretation, or the decision to submit the work for publication.

### Author contributions

Kangcheng Song, Conceptualization, Data curation, Formal analysis, Validation, Investigation, Visualization, Writing - original draft, Writing - review and editing; Miao Wei, Data curation, Formal analysis, Validation, Investigation, Visualization, Writing - original draft, Writing - review and editing; Wenjun Guo, Data curation, Validation, Investigation, Visualization, Writing - original draft, Writing - review and editing; Li Quan, Formal analysis, Investigation, Methodology, Writing - review and editing; Yunlu Kang, Jing-Xiang Wu, Data curation, Investigation; Lei Chen, Conceptualization, Resources, Data curation, Formal analysis, Supervision, Funding acquisition, Validation, Investigation, Visualization, Methodology, Writing - original draft, Project administration, Writing - review and editing

### Author ORCIDs

Kangcheng Song https://orcid.org/0000-0001-7932-2202
Miao Wei https://orcid.org/0000-0002-6304-4778
Jing-Xiang Wu http://orcid.org/0000-0001-9851-0065
Lei Chen https://orcid.org/0000-0002-7619-8311

Decision letter and Author response
Decision letter https://doi.org/10.7554/eLife.63429.sa1
Author response https://doi.org/10.7554/eLife.63429.sa2

## Additional files

### Supplementary files
• Transparent reporting form

### Data availability

The density maps for hTRPC5 have been deposited to the Electron Microscopy Data Bank (EMDB) under the accession number: EMD-30987 for apo hTRPC5, EMD-30575 for CMZ-bound hTRPC5, and EMD-30576 for HC-070-bound hTRPC5. Coordinates of the atomic model have been deposited in the Protein Data Bank (PDB) under accession number 7E4T for apo hTRPC5, 7D4P for CMZ-bound hTRPC5 and 7D4Q for HC-070-bound hTRPC5.

The following datasets were generated:

| Author(s) | Year | Dataset title | Dataset URL | Database and Identifier |
|---|---|---|---|---|
| Wei M,  Song K, Chen L | 2020 | cryo-EM structure of hTRPC5 in complex with Clemizole | https://www.ebi.ac.uk/pdbe/entry/emdb/EMD-30575 | Electron Microscopy Data Bank, EMD-30575 |
| Wei M,  Song K, Chen L | 2020 | cryo-EM structure of hTRPC5 in complex with Clemizole | https://www.rcsb.org/structure/7D4P | RCSB Protein Data Bank, 7D4P |
| Wei M,  Song K, Chen L | 2020 | cryo-EM structure of hTRPC5 in complex with HC-070 | https://www.ebi.ac.uk/pdbe/entry/emdb/EMD-30576 | Electron Microscopy Data Bank, EMD-30576 |
| Wei M,  Song K, Chen L | 2020 | cryo-EM structure of hTRPC5 in complex with HC-070 | https://www.rcsb.org/structure/7D4Q | RCSB Protein Data Bank, 7D4Q |
| Wei M,  Song K, Chen L | 2020 | cryo-EM structure of hTRPC5 in apo state | https://www.ebi.ac.uk/pdbe/entry/emdb/EMD-30987 | Electron Microscopy Data Bank, EMD-30987 |
| Wei M,  Song K, Chen L | 2020 | cryo-EM structure of hTRPC5 in apo state | https://www.rcsb.org/structure/7E4T | RCSB Protein Data Bank, 7E4T |

The following previously published dataset was used:

| Author(s) | Year | Dataset title | Dataset URL | Database and Identifier |
|---|---|---|---|---|
| Tang Q,  Guo W, Zheng L,  Wu JX, Liu M,  Zhou X, Zhang X,  Chen L | 2018 | Cryo-EM structure of human TRPC6 at 3.8A resolution | https://www.ebi.ac.uk/pdbe/entry/emdb/EMD-6856 | Electron Microscopy Data Bank, EMD-6856 |

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
