## [Decision Letter]

**Acceptance summary:**

The TRPC5 channels are pharmaceutical targets for diverse pathologies, including anxiety, depression and kidney disease. The current manuscript presents structures of the human TRPC5 channel in complex with two different, mechanistically distinct inhibitors, clemizole and HC-070. The structural work in this manuscript is well-executed, with high quality maps and clear density for the inhibitors. Structural observations are coupled with extensive mutagenesis experiments to dissect the contributions of binding site residues to channel inhibition and to identify the molecular determinants of inhibitor selectivity for TRPC5 over closely related TRP channels. This work provides important mechanistic insight into the pharmaceutical modulation of TRPC5.

**Decision letter after peer review:**

Thank you for submitting your article "Structural basis for human TRPC5 channel inhibition by two distinct inhibitors" for consideration by *eLife*. Your article has been reviewed by three peer reviewers, including Randy B Stockbridge as the Reviewing Editor and Reviewer #1, and the evaluation has been overseen by Richard Aldrich as the Senior Editor. The following individual involved in review of your submission has agreed to reveal their identity: Erhu Cao (Reviewer #2).

The reviewers have discussed the reviews with one another and the Reviewing Editor has drafted this decision to help you prepare a revised submission.

Summary:

In this article Song et al. present structures of the human TRPC5 channel in complex with inhibitors clemizole and HC-070 at 2.7 Å resolution. The authors identify the inhibitor binding sites and conduct site-directed mutagenesis studies in order to dissect the contribution of binding site residues to channel inhibition and to identify the molecular determinants of inhibitor selectivity for TRPC5 vs. TRPC3/6/7.

The structural work in this manuscript is generally well-executed, with high quality maps and clear density for the drugs. The authors perform extensive mutagenesis to identify positions that alter the IC_50_ of the drugs. However, the reviewers were disappointed that in many cases, intriguing findings were not followed up with experiments or analysis to lead to mechanistic insights. In addition, the drug binding sites are in regions of the protein that are more generally involved in channel gating. Thus, basic electrophysiological characterization of the mutants with altered IC_50_s is an important control to establish whether the effects of these mutations are, in fact, primarily on the affinity of inhibitor binding.

Essential revisions:

1) It is essential to understand whether the mutants that alter inhibition also affect channel function. This would be an important control, as mutating the residues could result in a shift in gating kinetics and/or surface expression that might be reflected as a change in IC_50_. For example, in the Discussion, the authors point out that residues from inhibitor binding pocket B contribute to channel gating. This underscores the importance of characterizing the mutant behavior to distinguish the effects of mutations on gating kinetics vs. inhibitor binding. In addition, Duan et al., 2019, found that mutations to residue F576 altered channel function. The authors mutate a number of residues in this region (L572, Q573, F576…) and find shifts in IC_50_ compared with wild type. However, without electrophysiological characterization of these mutants it is difficult to pin the shift in IC_50_ to a change in affinity for the inhibitor alone.

2) Likewise, it is important to understand whether intracellular Ca^2+^ is required for DMZ inhibition of TRPC5 since Icilin activates TRPM8 in a Ca^2+^-dependent manner and binds to a topologically equivalent site as CMZ.

3) It's not clear why the dose-response curves are plateau at values higher than 100%. In addition, from the Materials and methods, it looks like the Hill coefficient was allowed to vary in fitting the does response curves. Fits will improve if the number of parameters are increased, but if there is no mechanistic rationale, then this should not be done.

4) A number of interesting structural observations are made, but not translated into mechanistic insight. For example, N443, L496, and S495 do not appear to directly interact with CLZ, but affect the IC_50_. For the mutant L528A, HC-070 activates the channel in a dose-dependent manner. What is the mechanistic insight to be gained from these observations? The Zn2+ binding site is an important structural finding. Understanding its role in channel regulation would strengthen this manuscript.

5) Comparison to the apo mTRPC5 structure are made, but (as the authors acknowledge) hard to interpret since the homologue and membrane mimetic are different. It is not clear that the subtle changes shown in Figure 5 will affect the pore configuration. If it is technically feasible, the authors should solve a structure of apo hTRP5 in the same membrane mimetic as the drug-bound structures to allow for a direct comparison.

6) The resolution of the two maps is excellent but the atomic models could be improved. The clash score for both is rather high and both models contain a rather high number of poor rotamers. This should be an easy fix given the high resolution of the maps.

7) This manuscript would be more useful to the field if the Discussion was restructured to make a more detailed comparison with the many TRPC inhibitor bound structures (and published functional data). Such a Discussion would enhance the mechanistic insights that could be taken from these structures.

---

## [Author Response]

Essential revisions:1) It is essential to understand whether the mutants that alter inhibition also affect channel function. This would be an important control, as mutating the residues could result in a shift in gating kinetics and/or surface expression that might be reflected as a change in IC_50_. For example, in the Discussion, the authors point out that residues from inhibitor binding pocket B contribute to channel gating. This underscores the importance of characterizing the mutant behavior to distinguish the effects of mutations on gating kinetics vs. inhibitor binding. In addition, Duan et al., 2019, found that mutations to residue F576 altered channel function. The authors mutate a number of residues in this region (L572, Q573, F576…) and find shifts in IC50 compared with wild type. However, without electrophysiological characterization of these mutants it is difficult to pin the shift in IC_50_ to a change in affinity for the inhibitor alone.

We fully agree with reviewers on that the shift in IC_50_ is not sufficient to indicate the change in inhibitor affinity. We tried whole cell recordings for characterization of hTRPC5 mutants. Both EA and [Ca^2+^]_o_ can activate wild-type hTRPC5. But Duan et al., 2019, found mutations in IBP-B abolished EA activation. Therefore, EA activation is not suitable to characterize hTRPC5 mutants at HC-070 binding sites (IBP-B). So we tried whole cell recording of hTRPC5 currents activated by [Ca^2+^]_o_, with 5 mM EDTA and 5 mM EGTA in pipette solution.

**Author response image 1. sa2fig1:** 

In this condition, we found the wild-type hTRPC5 currents activated by [Ca^2+^]_o_ had very slow kinetics which were different from cell to cell, suggesting this is not suitable for robust characterization of mutants either.Alternatively, we performed FLIPR experiments to measure the dose-response activation of hTRPC5 mutants by [Ca^2+^]_o_ in Figure 2—figure supplement 2B-E. We have also determined the surface expression of these mutants in Figure 2—figure supplement 2F-G. These results collectively indicated that mutants showed robust surface expression and have similar trends of activation by [Ca^2+^]_o_. However, due to the unavailability of radioactive ligand binding assays, we cannot attribute the changes in IC_50_ to the changes in inhibitor affinity. For explicitness, we have changed "affinity" into "potency" throughout the manuscript.

2) Likewise, it is important to understand whether intracellular Ca^2+^ is required for DMZ inhibition of TRPC5 since Icilin activates TRPM8 in a Ca^2+^-dependent manner and binds to a topologically equivalent site as CMZ.

We have provided the electrophysiological data in Figure 4—figure supplement 1C. We used the whole cell mode with 1 mM EGTA/1 mM EDTA in pipette solution and 10 mM EGTA in bath solution to chelate calcium. We found pre-incubation of CMZ with the hTRPC5-expressing cell can inhibit the current evoked by EA, shown in revised Figure 4—figure supplement 1C, suggesting high intracellular Ca^2+^ is not absolutely required for CMZ inhibition. We indicate this in the Results.

3) It's not clear why the dose-response curves are plateau at values higher than 100%. In addition, from the Materials and methods, it looks like the Hill coefficient was allowed to vary in fitting the does response curves. Fits will improve if the number of parameters are increased, but if there is no mechanistic rationale, then this should not be done.

We have determined the calcium activation levels of wild type hTRPC5 and mutants in Figure 2—figure supplement 2A, and removed the analysis of mutants with response less than 50% of wild type hTRPC5 due to low signal-to-noise ratio. Moreover, we have also re-analyzed the FLIPR data using the averaged base line as background and re-plotted all of the dose-response curves in normalization to wild type hTRPC5 (see revised Materials and methods section). For curve fitting, we found if we fixed the Hill coefficient to 1, the fitted curve would markedly deviates from experimental data points (Author response image 2).

We reasoned that there is probably some cooperativity of inhibitor binding to four subunits of hTRPC5 channel, similar to what observed recently by others (PMID: 33230284). Therefore, we allowed Hill coefficient to change during curve fitting and provided the numbers obtained in revised Table 2 and Table 3.

4) A number of interesting structural observations are made, but not translated into mechanistic insight. For example, N443, L496, and S495 do not appear to directly interact with CLZ, but affect the IC_50_. For the mutant L528A, HC-070 activates the channel in a dose-dependent manner. What is the mechanistic insight to be gained from these observations? The Zn2+ binding site is an important structural finding. Understanding its role in channel regulation would strengthen this manuscript.

In Figure 2C-D, we have shown N443 is in close proximity to CMZ and probably has Van der Waals interactions with CMZ. Replacement of N443 with Leu might change these interactions. We have indicated this in the Results. L496A, L496M and L528A showed low signals (Figure 2—figure supplement 2A) and are not suitable for robust characterization and we have removed these mutants in the revised manuscript. We found the triple mutant C176A-C178A-C181A in zinc binding site retained reasonable surface expression and activation by both calcium and EA (Figure 2—figure supplement 2A, E-G). Therefore, we propose this zinc binding site might play other regulatory or structural roles, instead of ion channel gating, as indicated in the Results.

5) Comparison to the apo mTRPC5 structure are made, but (as the authors acknowledge) hard to interpret since the homologue and membrane mimetic are different. It is not clear that the subtle changes shown in Figure 5 will affect the pore configuration. If it is technically feasible, the authors should solve a structure of apo hTRP5 in the same membrane mimetic as the drug-bound structures to allow for a direct comparison.

We have provided the apo hTRPC5 structure in revised manuscript shown in Figure 1—figure supplements 2-3 for direct structural comparison in revised Figure 5.

6) The resolution of the two maps is excellent but the atomic models could be improved. The clash score for both is rather high and both models contain a rather high number of poor rotamers. This should be an easy fix given the high resolution of the maps.

We have further refined the structural model with new statistics shown in Table 1.

7) This manuscript would be more useful to the field if the Discussion was restructured to make a more detailed comparison with the many TRPC inhibitor bound structures (and published functional data). Such a Discussion would enhance the mechanistic insights that could be taken from these structures.

We have provided structural comparisons with new hTRPC4 and hTRPC5 inhibitor-bound structures in the Discussion and Figure 6—figure supplement 1.